# Metabolic Transition of Milk Triacylglycerol Synthesis in Response to Varying Levels of Three 18-Carbon Fatty Acids in Porcine Mammary Epithelial Cells

**DOI:** 10.3390/ijms22031294

**Published:** 2021-01-28

**Authors:** Yantao Lv, Fang Chen, Shihai Zhang, Jun Chen, Yinzhi Zhang, Min Tian, Wutai Guan

**Affiliations:** 1College of Animal Science and National Engineering Research Center for Breeding Swine Industry, South China Agricultural University, Guangzhou 510642, China; lvyantao0818@163.com (Y.L.); chenfang1111@scau.edu.cn (F.C.); zhangshihai@scau.edu.cn (S.Z.); chenjun7080@126.com (J.C.); jiyuji_02@163.com (Y.Z.); tianmin@stu.scau.edu.cn (M.T.); 2Innovative Institute of Animal Healthy Breeding, College of Animal Sciences and Technology, Zhongkai University of Agriculture and Engineering, Guangzhou 510225, China

**Keywords:** 18-carbon fatty acids, porcine mammary epithelial cells, milk fat biosynthesis, lipogenic genes

## Abstract

This study aimed to examine the effects of increasing levels of three 18-carbon fatty acids (stearate, oleate and linoleate) on mammary lipogenesis, and to evaluate their effects on the milk lipogenic pathway in porcine mammary epithelial cells (pMECs). We found that increasing the three of 18-carbon fatty acids enhanced the cellular lipid synthesis in a dose-dependent manner, as reflected by the increased (triacylglycerol) TAG content and cytosolic lipid droplets in pMECs. The increased lipid synthesis by the three 18-carbon fatty acids was probably caused by the up-regulated expression of major genes associated with milk fat biosynthesis, including *CD36* (long chain fatty acid uptake); *GPAM*, *AGPAT6*, *DGAT1* (TAG synthesis); *PLIN2* (lipid droplet formation); and *PPARγ* (regulation of transcription). Western blot analysis of CD36, DGAT1 and PPARγ proteins confirmed this increase with the increasing incubation of 18-carbon fatty acids. Interestingly, the mRNA expressions of *ACSL3* and *FABP3* (fatty acids intracellular activation and transport) were differentially affected by the three 18-carbon fatty acids. The cellular mRNA expressions of *ACSL3* and *FABP3* were increased by stearate, but were decreased by oleate or linoleate. However, the genes involved in fatty acid de novo synthesis (*ACACA* and *FASN*) and the regulation of transcription (*SREBP1*) were decreased by incubation with increasing concentrations of 18-carbon fatty acids. In conclusion, our findings provided evidence that 18-carbon fatty acids (stearate, oleate and linoleate) significantly increased cytosolic TAG accumulation in a dose-dependent manner, probably by promoting lipogenic genes and proteins that regulate the channeling of fatty acids towards milk TAG synthesis in pMECs.

## 1. Introduction

The mammary gland is a major lipid-synthesizing organ during lactation, and in sows, the lactating mammary gland is estimated to secrete approximately 8 kg of milk per day, with a fat content of approximately 5% [1]. Mammalian milk fat is composed of 98% triacylglycerols (TAG), which are composed of three fatty acids esterified to a three-carbon glycerol backbone [2]. The lactating porcine mammary gland synthesizes about 400 g of TAG daily, or nearly 8.4 kg fat during a lactation period of just 21 days, making it a formidable lipid-synthesizing and secreting organ. The three 18-carbon fatty acids, stearate, oleate and linoleate, were the major fatty acids in sow milk, contributing about 50% of the total fatty acids esterified to TAG. On average, the stearate, oleate and linoleate contents (percent of total fatty acids, wt. %) were 4.5, 32 and 14, respectively [3]. The long chain fatty acids (including 18-carbon fatty acids) in mammary gland have two main origins: (1) uptake from TAG carried in chylomicrons and very low-density lipoproteins by lipoprotein lipase (LPL), and from adipose tissue by hormone-sensitive lipase (HSL); (2) de novo synthesis by mammary epithelial cell [4,5].

The exogenous fatty acid supplementation represents an important way to modulate the milk fat content and fatty acid composition in lactating sows, because the lactating mammary lipogenesis is highly responsive to dietary fatty acids’ concentration and composition [6]. For example, the addition of 7.5–15% fat to the sows’ diets during late gestation and (or) lactation increases milk production and fat content in colostrum and milk [7]. The addition of a fat source of 18-carbon fatty acids (e.g., 8% animal fat rich in stearic acid, rapeseed oil rich in oleic acid or sunflower oil rich in linoleic acid) to the sow’s diet during lactation can increase the milk fat concentration and the accumulation of 18-carbon fatty acids [8].

The two transcription factors, including sterol-regulatory element binding protein 1 (SREBP 1) and peroxisome proliferator-activated receptor γ (PPARγ), play a key role in the regulation of milk fat synthesis in lactating animals [3,9,10]. SREBP1 preferentially promotes the expression of fatty acid biosynthesis-related genes, such as acetyl-CoA carboxylase α (ACACA) and fatty acid synthase (FASN), and its expression level in mammary gland tissue has thus been associated with milk fat content and milk fatty acid composition [9,11,12]. PPARγ is thought to play a role in regulating milk fat synthesis because PPARγ, along with its lipogenic target genes, is up-regulated in lactating mammary tissue [3,9,10]. The activation of PPARγ by its agonist rosiglitazone (ROSI) resulted in a marked increase in the expression of genes associated with TAG synthesis and secretion in goat mammary epithelial cells [13]. 

Given that milk fat concentration and fatty acid composition are closely related to the content of dietary fat, and that the three 18-carbon fatty acids (stearate, oleate and linoleate) were the major fatty acids in sow milk fat, it is important to identify lipogenic genes that are sensitively regulated by the three 18-carbon fatty acids. In sows, however, there are few reports in the literature available on the role of 18-carbon fatty acid in regulating milk TAG synthesis. 

Mammary epithelial cells are well known as a cell type responsible for TAG synthesis and secretion. Therefore, in this study, by using porcine mammary epithelial cells (pMECs) as a model, we aimed to investigate the molecular mechanism by which different levels of 18-carbon fatty acids (stearate, oleate and linoleate) regulate lipid synthesis in pMECs. The results of this study provide insights into the physiological functions of stearate, oleate and linoleate in milk lipid synthesis at the cellular level.

## 2. Results

### 2.1. Influencing Cell Viability of pMECs

Both oleate and linoleate significantly increased the viability of pMECs in a dose-dependent manner (50–600 μM) (*p* < 0.05, Figure 1B,C). However, the incubation of cells with 25–50 μM stearate for 24 h did not affect the cell viability, but exposure to 100–600 μM stearate decreased the cell viability approximately by 31–44% (Figure 1A). It is indicated that unsaturated 18-carbon LCFAs (oleate and linoleate) promoted cell viability in a dose-dependent manner (50–600 μM), whilst saturated 18-carbon LCFA (100–600 μM stearate) decreased cell viability in pMECs.

### 2.2. Enhancing Accumulation of Intracellular TAG

Addition of the three 18-carbon fatty acids (stearate, oleate or linoleate) in the medium for 24 h significantly increased cellular TAG contents in a dose-dependent manner (Figure 2). Similarly, Oil Red O staining (Figure 3) further confirmed the enhanced formation of cytosolic lipid droplets in pMECs when incubated with increasing levels of stearate, oleate or linoleate. The average diameter of large lipid droplets was increased linearly or quadratically with increasing stearate, oleate or linoleate (*p* < 0.05) (Figure 3). These results indicate that the three 18-carbon fatty acids (stearate, oleate or linoleate) increased cytosolic TAG accumulation and the formation of lipid droplets in pMECs in a dose-dependent manner.

### 2.3. Influencing Expression of Genes or Proteins Associated with LCFA Uptake, Intracellular Activation and Transport in pMECs

The effects of graded concentrations of 18-carbon fatty acids on mRNA expression of genes involved in lipid synthesis in pMECs are summarized in Table 1, Table 2 and Table 3. The heatmap (Figure 4) was created by clustering genes related to lipid synthesis in response to graded concentrations of stearate, oleate or linoleate. This showed the genes in several biological processes, including fatty acid uptake, import into cells, fatty acid activation, intra-cellular transport, fatty acid de novo synthesis and desaturation, TAG synthesis, lipid droplet formation and regulation of transcription.

Incubation with 18-carbon fatty acids (stearate, oleate or linoleate) for 24 h dose-dependently up-regulated the mRNA expression of *CD36* associated with LCFA uptake (Table 1, Table 2 and Table 3). Particularly, *CD36* mRNA expression was increased linearly or quadratically with increasing stearate, oleate or linoleate, with maximum at 400 μM stearate, 600 μM oleate or linoleate, respectively (*p* < 0.05, Table 1, Table 2 and Table 3). Consistent with its gene mRNA expression, cellular CD36 protein expression was significantly up-regulated by 100–600 μM 18-carbon fatty acids (stearate, oleate and linoleate) for 24 h (*p* < 0.05, Figure 5). Incubation with 25–600 μM 18-carbon fatty acids for 24 h differentially influenced the expression of genes associated with intracellular activation (*ACSL3*) and transport (*FABP3*) (Table 1). Cellular mRNA expressions of *ACSL3* and *FABP3* were increased linearly or quadratically with increasing stearate (*p* < 0.05, Table 1), whereas *ACSL3* and *FABP3* mRNA expression were decreased linearly or quadratically with 100~600 μM oleate or linoleate (*p* < 0.05, Table 2 and Table 3).

### 2.4. Down-Regulating the Expression of Genes or Proteins Related to FA De Novo Synthesis in pMECs

Incubation with 18-carbon fatty acids (stearate, oleate or linoleate) for 24 h suppressed the expression of genes associated with FA de novo synthesis (*ACACA* and *FASN*). *ACACA* and *FASN* mRNA expression in pMECs were decreased linearly or quadratically with increasing stearate, oleate or linoleate (*p* < 0.05, Table 1, Table 2 and Table 3). The transcript abundance of *ACACA* in pMECs treated with 100–600 μM linoleate was 28–40% lower than control, and that treated with 50–600 μM stearate was 23–58% lower than control, while oleate has no effects on the mRNA expression of ACACA. Incubation with stearate and oleate for 24 h changed the protein expression of ACACA. Compared with the control, the protein expression of ACACA in pMECs was increased 0.5–1.5-fold by 25–100 μM stearate (*p* < 0.05), but was not affected by a higher concentration of stearate (Figure 4A). Compared with the control, the ACACA protein expression in pMECs was increased –70% by 400–600 μM oleate (*p* < 0.05, Figure 5B,C). Linoleate did not affect the expression of cellular ACACA at either the mRNA or protein level.

The mRNA abundance of *FASN* in pMECs was decreased linearly or quadratically with 25–600 μM stearate, oleate or linoleate, respectively. The mRNA expressions of *SCD* were increased by 400–600 μM stearate (*p* < 0.05, Table 1), and this increase reached twofold with 400 μM stearate.

### 2.5. Influencing the Expression of Genes or Proteins Related to TAG Synthesis and Lipid Droplet Formation in pMECs

Incubation with the three 18-carbon fatty acids (stearate, oleate or linoleate) for 24 h enhanced the mRNA expression of genes associated with TAG synthesis (*GPAM*, *AGPAT6*, *LPIN2*, *DGAT1*) and lipid droplet formation (*PLIN2*). The cellular mRNA expressions of *GPAM*, *AGPAT6*, *LPIN2*, *DGAT1* and *PLIN2* were increased linearly or quadratically with increasing stearate (*p* < 0.05, Table 1, Table 2 and Table 3). Consistent with its gene mRNA expression, DGAT1 protein expression in pMECs was significantly up-regulated by 50–600 μM stearate, with the highest values observed at 400 μM stearate (*p* < 0.05, Figure 5A). Similarly, the cellular mRNA expressions of *GPAM*, *AGPAT6*, *LPIN1* or *2*, *DGAT1* and *PLIN2* were increased linearly or quadratically with increasing oleate or linoleate, respectively (*p* < 0.05, Table 2 and Table 3). *PLIN2* expression was significantly increased by oleate or linoleate, and its expression increased 16.5-fold and 14.1-fold to peak values at 600 μM oleate and 400 μM linoleate, respectively. DGAT1 protein expression was significantly increased 8.0-fold and 7.4-fold (*p* < 0.05) with 600 μM oleate and linoleate, respectively, compared to the control.

### 2.6. Influencing the Expression of Genes or Proteins Related to Regulation of Transcription in pMECs

Incubation with the three 18-carbon fatty acids (stearate, oleate or linoleate) for 24 h changed the mRNA expression of genes associated with the regulation of transcription (*SREBP1* and *PPARγ*, Table 1, Table 2 and Table 3). *PPARγ* mRNA expressions were increased linearly or quadratically (*p* < 0.05) with increasing stearate, oleate or linoleate. Consistent with its gene mRNA expression, cellular PPARγ protein expression was significantly up-regulated by the 18-carbon fatty acids (stearate, oleate and linoleate) (*p* < 0.05, Figure 5). Conversely, the mRNA expressions of *SREBP1* were decreased linearly or quadratically with increasing stearate, oleate or linoleate (*p* < 0.05, Table 1–3), respectively. Consistent with its gene mRNA expression, cellular SREBP1 protein expression was decreased or tended to be decreased by increasing 18-carbon fatty acids (stearate, oleate and linoleate) for 24 h (Figure 5B).

The transcript abundances of other transcription factors, such as *SCAP*, *INSIG1* or *PPARα*, were affected by increasing 18-carbon fatty acids in pMECs. The mRNA expressions of both *SCAP* and *PPARα* were decreased linearly or quadratically with increasing stearate (*p* < 0.05, Table 1). *INSIG1* mRNA expressions in pMECs were decreased by 50–100 μM stearate, but increased by 400–600 μM stearate (*p* < 0.05). The mRNA expressions of both *SCAP* and *INSIG1* were decreased linearly or quadratically with increasing oleate (*p* < 0.05, Table 2). *PPARα* mRNA expressions in pMECs were not affected by oleate (Table 2). The mRNA expression of *INSIG1* was decreased linearly or quadratically with increasing linoleate (*p* < 0.05, Table 3). *PPARα* and *SCAP* mRNA expressions in pMECs were not affected by linoleate (Table 3).

## 3. Discussion

In this study, we found that ≤50 μM stearate did not affect the viability of pMECs, but higher concentrations (100–600 μM) of stearate decreased the cell viability, while both oleate and linoleate significantly increased pMECs cell viability in a dose-dependent manner (50–600 μM) (*p* < 0.05) (Figure 1). Similar to our results, it was reported in bovine mammary epithelial cells (bMECs) that cell proliferation was suppressed by 100 μM stearate, while 100 μM oleate or linoleate enhanced cell viability [14]. It can be assumed that a high concentration of saturated 18-carbon LCFAs (stearate) inhibited the cell viability, while unsaturated 18-carbon LCFAs (oleate or linoleate) promoted cell viability. The mechanism by which unsaturated 18-carbon LCFAs (oleate and linoleate) promoted cell proliferation and survival is probably involved in the activation of the phosphorylation of extracellular signal-regulated kinase (ERK) 1/2 and Akt kinase [14]. Whilst the mechanism is probably involved in increased cytochrome c release caused by reduced mitochondrial membrane potential, some studies have been performed on palmitate-induced apoptosis by which saturated LCFAs induce apoptosis [15]. It is notable that the TAG accumulation was enhanced by 100–600 μM stearate, even when the cell viability was suppressed by these concentrations of stearate. 

In this study, we also found that the addition of exogenous stearate, oleate or linoleate significantly increased cellular TAG contents in a dose-dependent manner, and the increased accumulation of TAG content is probably associated with the enhanced formation of lipid droplets, as larger droplets and up-regulated mRNA expression related to droplet formation were observed in the treatments of exogenous 18-carbon fatty acids. Cytoplasmic lipid droplets are the immediate precursors of milk lipids, and function as the place where newly formed lipids and other neutral lipids are stored [16]. 

In mammary epithelial cells, the fatty acids used for lipid synthesis are derived from direct uptake from blood, or are synthesized de novo from substrates such as butyrate and acetate. Both of the pathways were evaluated in this study, but CD36 has been reported to be the key transporter for exogenous LCFA trans-membrane transport in lactating mammary gland [3,9,10]. In this study, the increased mRNA and protein expression for CD36 by stearate, oleate or linoleate indicates that the exogenous 18-carbon fatty acid provision to mammary epithelial cells is able to activate the intracellular LCFA uptake. Consistent with our results, it was reported in bMECs that 18-carbon fatty acids (stearate, oleate or linoleate) increased the exogenous fatty acid transportation by up-regulating cellular *CD36* mRNA expression [17,18,19]. 

FASN and ACACA are considered the crucial enzymes of cellular fatty acid de novo synthesis in the mammary gland, which have been reported to be the main sources of short- and medium-chain fatty acids (almost all C4:0~C14:0 and approximately 50% of palmitic acid) in milk [6,20]. In this study, the fatty acid de novo synthesis in pMECs was suppressed by the three 18-carbon fatty acids (stearate, oleate or linoleate), as reflected by the down-regulated genes for *ACACA* and *FASN*. This is in agreement with previous works in bMECs, showing that treatment with stearate, oleate or linoleate decreased the cellular mRNA expression of *ACACA* and *FASN* [17]. These results indicate that exogenous 18-carbon LCFA, including stearate, oleate or linoleate, could suppress the fatty acid de novo synthesis in pMECs. De novo lipogenesis has been shown to be highly responsive to changes in the dietary composition [21]. However, animal studies have shown in mice that a high-fat diet suppresses hepatic de novo lipogenesis [22]. Similarly, the addition of 15 g/d conjugated linoleic acids (CLA) to a ewe’s lactation diet can suppress fatty acid de novo synthesis-related genes (*ACACA* and *FASN*) in the mammary tissue [23]. In contrast to dietary fat, high-carbohydrate diets induced an increase in hepatic de novo lipogenesis [24]. Interestingly, the type of dietetic carbohydrate affects the rate of fatty acid de novo synthesis, and simple sugars are more effective than complex carbohydrates in stimulating hepatic de novo lipogenesis [25,26]. Short-chain fatty acids also affect de novo lipogenesis, for instance, β-hydroxybutyrate (BHBA) can increase the expression of ACACA and FASN to increase milk fat synthesis in bMECs [27]. It is indicated that the source of energy substrate can differentially affect the fatty acid de novo synthesis. 

Once the LCFA enter cells, LCFA is activated by ACSLs to bind an acyl coenzyme A (CoA), then the resulting acyl-CoAs have numerous metabolic fates within cells, including incorporation into TAG and membrane phospholipids, used as substrates for β-oxidation and protein acylation, and function as ligands for transcription factors [28]. FABPs are members of the superfamily of lipid-binding proteins (LBP) that regulate FA uptake and intracellular transport [29]. FABP facilitates the cytosolic transport of both long-chain saturated and unsaturated fatty acids. The ACSL isoforms differ in their substrate preferences, enzyme kinetics and intracellular locations, and then each ACSL isoform channels FA toward separate metabolic fates [30]. In the current study, cellular mRNA expressions of *ACSL3* and *FABP3* in pMECs were increased by stearate (*p* < 0.05, Table 1), but decreased by oleate or linoleate (*p* < 0.05, Table 2 and Table 3). These results are in accordance with the previous results in bMECs [17]. It therefore can be speculated that the effect of exogenous LCFA on the activation and intracellular transport of fatty acid may be associated with fatty acid saturation. Notably, the lipid synthesis in pMECs was enhanced even though the activation and intracellular transport was suppressed by oleate or linoleate.

In most mammalian tissues, including the mammary gland, the majority of TAG is synthesized at the endoplasmic reticulum (ER) membrane through the glycerol 3-phosphate pathway, which involves the esterification of fatty acids to a glycerol 3-phosphate backbone [31]. The glycerol 3-phosphate pathway is regulated by the sequential action of glycerol-3-phosphate acyltransferases (GPATs), 1-acylglycerol-3-phosphate acyltransferases (AGPATs), lipin phosphatidic acid phosphatase (PAP) proteins, and diacylglycerol acyltransferases (DGATs) [32]. PLIN2 (adipophilin), which locates on the droplet surface, is associate with lipid droplet storage and the control of cellular lipolytic activity [33,34]. The transcripts of *AGPAT1*, *LPIN1/2*, *DGAT1* and *PLIN2* are the most abundant within each specific gene family in the lactating porcine mammary gland, and the mRNA expressions of these genes were up-regulated during lactation [3]. In this study, the mRNA expressions of genes associated with TAG synthesis (*GPAM*, *AGPAT1/6*, *LPIN1/2*, *DGAT1*) and lipid droplet formation (*PLIN2*) in pMECs were up-regulated by 18-carbon fatty acids (stearate, oleate or linoleate). This is in accordance with the promotive effect of 18-carbon fatty acids on the cellular TAG and lipid droplets formation. It can be assumed, based on these results, that TAG synthesis and lipid formation in MECs are enhanced by exogenous 18-carbon fatty acids (stearate, oleate or linoleate).

SREBP1 and PPARγ are of importance in the transcriptional regulation of many genes related to milk fat synthesis and secretion, and therefore control fatty acid synthesis and uptake and TAG synthesis in mammary cells [13,17,35,36]. In this study, all of the three 18-carbon fatty acids (stearate, oleate or linoleate) could increase *PPARγ* mRNA expression in pMECs, which indicates that 18-carbon fatty acids regulate TAG synthesis probably through activating *PPARγ* and their target lipogenic genes. However, 18-carbon fatty acids decreased the cellular mRNA expression of *SREBP1* in pMECs, which is in accordance with the report showing that 100 μM LCFA, including *cis*-9 18:1, *trans*-10 18:1, *trans*-10, and *cis*-12 18:2 and 20:5, down-regulated the expression of *SREBP1* in bMECs [17]. This indicates that 18-carbon fatty acids inhibit fatty acid de novo synthesis-related genes (ACACA and FASN), probably via regulation of SREBP1, since SREBP1 is a key regulator for up-regulating genes that encode proteins (ACACA and FASN) involved in fatty acid de novo synthesis in mammary epithelial cells [37]. Most LCFAs, including 18-carbon fatty acids, and specifically polyunsaturated fatty acids (PUFA), which are natural ligands and bind to PPARγ, can elicit changes in gene expression and rates of lipogenesis [38,39]. It is indicated that the mammary epithelial cells prefer to utilize exogenous LCFA rather than de novo synthesized fatty acids for lipid synthesis when mammary cells have access to exogenous LCFA. This is probably because the utilization of exogenous fatty acids may represent a convenient means of lipid synthesis, while the fatty acid de novo synthesis is an energy- and time-consuming process. Based on our results, we concluded that when exogenous palmitate is provided in the culture media at physiological concentrations, the uptake of extracellular LCFA plays a major role in enhanced TAG synthesis and lipid formation in pMECs, while fatty acid de novo synthesis accounts for a minor fraction of intracellular TAG.

The development of the modern sow has resulted in an animal with less body fat reserves and lesser appetite [40]. As the demands for milk and nutrient output have increased substantially for larger and fast-growing litters, the catabolism of maternal reserves commonly occurs in lactating sows to ensure milk output due to limited energy intake [41,42]. So, the dietary energy source should be formulated to support a high level of milk fat production, prevent sow’s tissue mobilization, and maximize long-term productivity. Indeed, dietary supplementation with optimal amounts of lipid to sows is an effective way to increase the milk fat by protecting both pathways [43]. Given that mammary epithelial cells prefer exogenous LCFA in synthesizing TAG, the dietary addition of optimal amounts of fat to support lipid synthesis from two origins may represent the most efficient means of promoting milk fat synthesis. Lipid supplementation increases average daily energy intake, which is partitioned for lactation, as indicated by greater milk fat output and improved litter growth rate [40]. In practical production, the addition of 3–5% fat to the sow’s diet during late pregnancy and lactation can effectively increase fat and energy output in sow milk, and improve growth performance of nursing piglets [44,45,46], increase the piglets survival [43], and improve the subsequent reproductive performance of sows [47]. Notably, the fat source (saturated vs. unsaturated; number of carbon) should also be considered in practical production.

## 4. Materials and Methods 

### 4.1. Cell Culture and Treatments

The pMECs were isolated, purified, and cultured from the mammary gland of a 17-day lactating Large White sow according to our previously described protocol [48]. Isolated cells were maintained in a basal Dulbecco’s modified Eagle’s medium/F12 (DMEM/F12) (Gibco, Grand Island, NY, USA), supplemented with 10% fetal bovine serum (FBS) (Gibco, Grand Island, NY, USA), Insulin-Transferrin Selenium (ITS) (5 μg/mL; ScienCell, Carlsbad, CA, USA), epidermal growth factor (EGF) (10 ng/mL; Gibco, Grand Island, NY, USA), IGF-1 (10 ng/mL; Sigma-Aldrich, St. Louis, MO, USA), hydrocortisone (5 μg/mL; Sigma-Aldrich, St. Louis, MO, USA), penicillin (100 U/mL) and streptomycin (100 μg/mL; Gibco, Grand Island, NY, USA). Cells were incubated at 37 °C in a humidified atmosphere with 5% CO_2_ and the culture medium was changed every 24 h. Fluorescence-activated cell sorting (FACS) analysis for cytokeratin expression in the cells revealed that they were composed of 90% mammary epithelial cells. Additionally, the cells had a high mRNA abundance of β-casein, determined using RT-PCR. In this study, pMECs from the 11th passages were used.

Once the cells were cultured reaching 80–90% confluence, the cells were washed twice with PBS (Gibco, Grand Island, NY, USA) and then the growth medium was replaced with complete DMEM/F12 in the presence of different concentrations of stearate, oleate or linoleate (0, 25, 50, 100, 200, 400, and 600 μM) for 24 h. Control cells were incubated with fresh medium (DMEM/F12) without 18-carbon fatty acids as well. Stearate (≥99% pure isomers), oleate (≥99% pure isomers) and linoleate (≥98% pure isomers) (Sigma-Aldrich, St. Louis, MO, USA) were conjugated to fatty acid-free bovine serum albumin (BSA) (Equitech-Bio, Kerrville, TX, USA) at a 4:1 ratio. 

### 4.2. Cell Viability Assay

The effects of stearate, oleate and linoleate on cell viability were tested via MTT assay. Briefly, after treatment with stearate, oleate or linoleate, the pMECs culture medium was removed and exchanged for a fresh one. The cells were incubated with 20 μL MTT (Sigma-Aldrich, St. Louis, MO, USA) solution (5 mg/mL PBS) at 37 °C for 4 h. The top medium MTT was then removed and 200 μL dimethyl sulfoxide (DMSO, Sigma-Aldrich, St. Louis, MO, USA) was added to each well. After 10 min, the absorbance of each well was measured with a multifunctional plate reader (SpectraMax M5, San Jose, CA, USA) at a wavelength of 490 nm. All assays were performed in triplicate. 

### 4.3. Assessment of Triglyceride Storage

The intracellular lipid accumulation was measured through Oil Red O staining. Briefly, after treatment with stearate, oleate or linoleate, the cells were washed with PBS twice, fixed in 4% paraformaldehyde (Sigma-Aldrich, St. Louis, MO, USA) for 30 min at room temperature and then rinsed with PBS three times (10 min each time). A 0.5% Oil Red O/isopropyl alcohol solution (Sigma-Aldrich, St. Louis, MO, USA) was added for 1 h to the cells, and then they were washed three times with PBS. The stained cytoplasmic lipids were visualized and photographed by an inverted microscope at 400× magnification. Lipid droplet diameter was measured using Image J software (NIH, Bethesda, MD, USA). In each field captured on camera, the mean diameter of the five largest lipid droplets was calculated and used to estimate the maximum diameter of the intracellular lipid droplet.

TAG content was measured via a TAG assay kit (Applygen, Beijing, China) according to the manufacturer’s instructions. Briefly, after treatment with stearate, oleate or linoleate, the culture medium was removed, and the remaining cells were washed with PBS and collected. The cell samples were treated with RIPA lysis buffer (Beyotime, Nanjing, China), and cell lysates were extracted and stored at −80 °C until analysis. TAG contents in supernatant were assayed using commercial kits (Applygen, Beijing, China). Protein concentrations in supernatant were determined using a Pierce BCA protein Assay kit (Thermo Fisher Scientific, Waltham, MA, USA). The content of TAG was determined by normalization to the total protein of each sample.

### 4.4. RNA Extraction and Real-Time Quantitative PCR

After treatment with stearate, oleate or linoleate, pMECs were washed twice with PBS. Total RNA was isolated from pMECs using TRIzol reagent (Invitrogen, Carlsbad, CA, USA) and treated with DNaseI (Takara, Tokyo, Japan) for removing DNA contamination. The purity of RNA (A260/A280) for all samples was 1.8–2.0 determined via a spectrophotometer (Thermo Fisher Scientific, Waltham, MA, USA), indicating that they were pure and clean, and the integrity of the RNA was also checked by ethidium bromide-stained agarose gel electrophoresis. The first strand of cDNA was reverse-transcribed from 1 μg of total RNA using a PrimeScript RT reagent kit with gDNA eraser (Takara, Tokyo, Japan). cDNA was then diluted 1:5 with DNase/RNase free water.

Primers were designed based on cDNA sequence (Appendix A) using Primer Premier 5 (PREMIER Biosoft Int., Palo Alto, CA, USA). The primers for target and reference genes are shown in Appendix A. The transcript abundances of target and reference genes were determined by *q*PCR. *q*PCR was performed with SYBR Green Real-Time PCR Master Mix (Toyobo, Tokyo, Japan) according to the manufacturer’s instructions. *q*PCR was run on an ABI Prism 7500 Sequence Detection System (Thermo Fisher Scientific, Waltham, MA, USA) using a SYBR^®^ PCR protocol. The PCR protocol was composed of an initial denaturation at 95 °C for 1 min, and 40 cycles of amplification comprising denaturation at 95 °C for 15 s, annealing at primer-specific temperatures (58–61 °C) for 15 s and elongation at 72 °C for 20 s. Melting curve analysis and the *q*PCR products were monitored using 1.5% agarose gel electrophoresis with ethidium bromide to evaluate the amplification specificity. mRNA levels of all samples were normalized to the values of the reference gene (*β-actin*) and the results were expressed as fold changes of the threshold cycle (Ct) value relative to the control using the 2^−ΔΔCt^ method [49]. 

### 4.5. Western Blot Analysis

After treatment with stearate, oleate or linoleate, cells were collected and total proteins were lysed using RIPA lysis buffer (Beyotime, Nanjing, China). The homogenates were combined with equal volumes of SDS sample buffer, and the proteins were separated by electrophoresis on a 5~12% polyacrylamide gel and transferred onto nitrocellulose membranes. The membranes were blocked with 5% skim milk in Tris-buffered saline with Tween-20, followed by overnight probing with the following primary antibodies: (1) CD36 (N-15) antibody (sc-5522, 1:500, Santa Cruz Biotechnology, Santa Cruz, CA, USA), (2) ACACA (T-18) antibody (sc-26817, 1:500, Santa Cruz Biotechnology, Santa Cruz, CA, USA), (3) DGAT1 antibody (ab59034, 1:500, Abcam, Cambridge, MA, USA), (4) SREBP1 (C-20) (sc-366, 1:500, Santa Cruz Biotechnology, Santa Cruz, CA, USA), (5) PPARγ (T-18) antibody (ab19481, 1:500, Abcam, Cambridge, MA, USA), (6) β-actin (C4) antibody (sc-47778, 1:1000, Santa Cruz Biotechnology, Santa Cruz, CA, USA). β-actin was intended to serve as a loading (internal) control. After washing, membranes were incubated with secondary antibody (ABR, Golden, CO, USA) and conjugated to HRP. The chemiluminescent signal was detected using ECL reagents (Beyotime, Nanjing, China) and bands were quantified by image processing software (Image Pro Plus 6.0, Media Cybernetics, Rockville, MD, USA).

### 4.6. Statistical Analysis

Data were analyzed using the general linear model procedure of the SAS software (SAS 9.0) as a completely randomized design. Polynomial contrasts were used to evaluate the linear and quadratic effects of stearate, oleate and linoleate on the various response criteria. Differences at *p* < 0.05 were considered statistically significant. Values are expressed as means ± SD. 

## 5. Conclusions

In summary, the results of the present study demonstrate that the three of 18-carbon fatty acids (stearate, oleate and linoleate) significantly increased cytosolic TAG accumulation in a dose-dependent manner. This is probably because stearate, oleate and linoleate can increase milk TAG synthesis in pMECs through activating the PPARγ pathway and then up-regulating the target genes associated with milk fat biosynthesis, including CD36 (LCFA uptake); GPAM, AGPAT6, DGAT1 (TAG synthesis); PLIN2 (lipid droplet formation) (Figure 6). Differentially, the mRNA expressions of ACSL3 and FABP3 (intracellular activation and transport) were increased by stearate but were decreased by oleate or linoleate. Furthermore, stearate, oleate and linoleate suppress milk fatty acid de novo synthesis through suppressing *ACACA* and *FASN* genes (fatty acid de novo synthesis) expression.

## Figures and Tables

**Figure 1 ijms-22-01294-f001:**
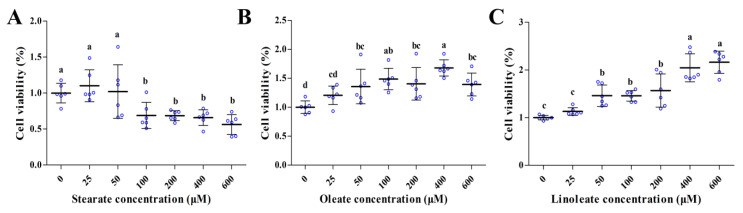
Effect of stearate, oleate and linoleate on cell viability in pMECs. pMEC cells were incubated with 0 (control), 25, 50, 100, 200, 400 and 600 μM stearate (**A**), oleate (**B**) or linoleate (**C**), respectively, for 24 h. Cell viability was estimated by MTT test. Values, expressed as percentage of control, are presented as mean ± SD (*n* = 6, blue dots display the number of measurements). Comparisons between groups were performed via ANOVA and Tukey’s range test for multiple comparisons. Different letters indicate statistical significance between different concentrations of stearate, oleate and linoleate treatment groups (*p* < 0.05).

**Figure 2 ijms-22-01294-f002:**
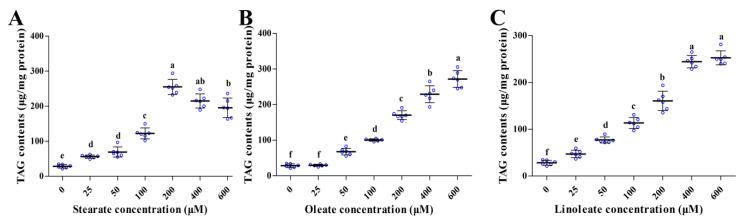
Effect of stearate, oleate and linoleate on cellular TAG content in pMECs. pMEC cells were incubated with 0 (control), 25, 50, 100, 200, 400 and 600 μM stearate (**A**), oleate (**B**) or linoleate (**C**), respectively, for 24 h. The data are expressed as the mean ± SD (*n* = 6, blue dots display the number of measurements). Comparisons between groups were performed via ANOVA and Tukey’s range test for multiple comparisons. Different letters indicate statistical significance between different concentrations of stearate, oleate and linoleate treatment groups (*p* < 0.05).

**Figure 3 ijms-22-01294-f003:**
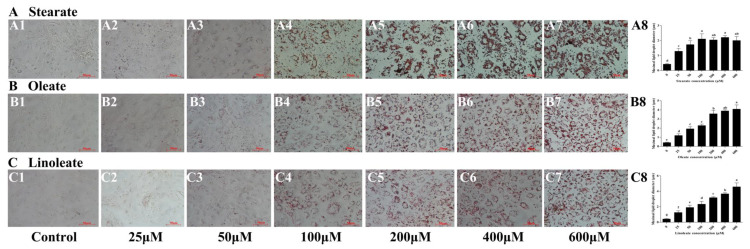
Effect of stearate, oleate and linoleate on lipid droplet formation in pMECs. Cells were incubated with 0 (control), 25, 50, 100, 200, 400 and 600 μM stearate (**A**), oleate **(B**) or linoleate (**C**), respectively, for 24 h, and then stained with oil red O and visualized by light microscopy with 400 × magnification. (A1), 0 μM stearate (Control); (A2), 25 μM stearate; (A3), 50 μM stearate; (A4), 100 μM stearate; (A5), 200 μM stearate; (A6), 400 μM stearate; (A7), 600 μM stearate; (A8), maximal lipid droplet diameter in the cells with 0 (control), 25, 50, 100, 200, 400 and 600 μM stearate; (B1), 0 μM oleate (Control); (B2), 25 μM oleate; (B3), 50 μM oleate; (B4), 100 μM oleate; (B5), 200 μM oleate; (B6), 400 μM oleate; (B7), 600 μM oleate; (B8), maximal lipid droplet diameter in the cells with 0 (control), 25, 50, 100, 200, 400 and 600 μM oleate; (C1), 0 μM linoleate (Control); (C2), 25 μM linoleate; (C3), 50 μM linoleate; (C4), 100 μM linoleate; (C5), 200 μM linoleate; (C6), 400 μM linoleate; (C7), 600 μM linoleate; (C8), maximal lipid droplet diameter in the cells with 0 (control), 25, 50, 100, 200, 400 and 600 μM linoleate. In A8, B8 and C8, data are expressed as mean ± SD (*n* = 5), and comparisons between groups were performed via ANOVA and Tukey’s range test for multiple comparisons. Different letters indicate statistical significance between different concentrations of stearate, oleate and linoleate treatment groups (*p* < 0.05).

**Figure 4 ijms-22-01294-f004:**
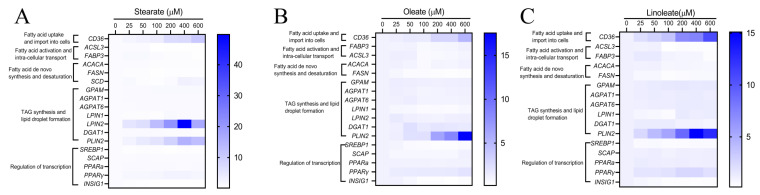
Representation of *q*PCR experimental data on mRNA expression of genes involved in lipid synthesis in pMECs. pMEC cells were incubated with 0 (control), 25, 50, 100, 200, 400 and 600 μM stearate (**A**), oleate (**B**) and linoleate (**C**), respectively, for 24 h. (**A**) Heatmap showing the expression levels of genes in response to graded-stearate concentration. (**B**) Heatmap showing the expression levels of genes in response to graded-oleate concentration. (**C**) Heatmap showing the expression levels of genes in response to graded-linoleate concentration.

**Figure 5 ijms-22-01294-f005:**
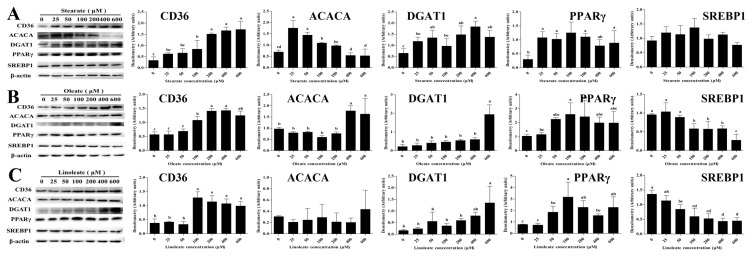
Effect of stearate, oleate and linoleate on the expression of proteins involved in lipid synthesis in pMECs. pMEC cells were incubated with 0 (control), 25, 50, 100, 200, 400 and 600 μM stearate (**A**), oleate (**B**) and linoleate (**C**), respectively, for 24 h. The proteins were separated by electrophoresis on a 5~12% polyacrylamide gel and transferred to nitrocellulose membranes, then the chemiluminescent signal was detected by using ECL reagents and bands were quantified by image processing software (Image Pro Plus 6.0). The expression levels of CD36, ACACA, DGAT1, PPARγ, and SREBP1 were normalized by that of β-actin. The data are expressed as the mean ± SD (*n* = 3). Comparisons between groups were performed via ANOVA and Tukey’s range test for multiple comparisons. Different letters indicate statistical significance between different concentrations of stearate, oleate and linoleate treatment groups (*p* < 0.05).

**Figure 6 ijms-22-01294-f006:**
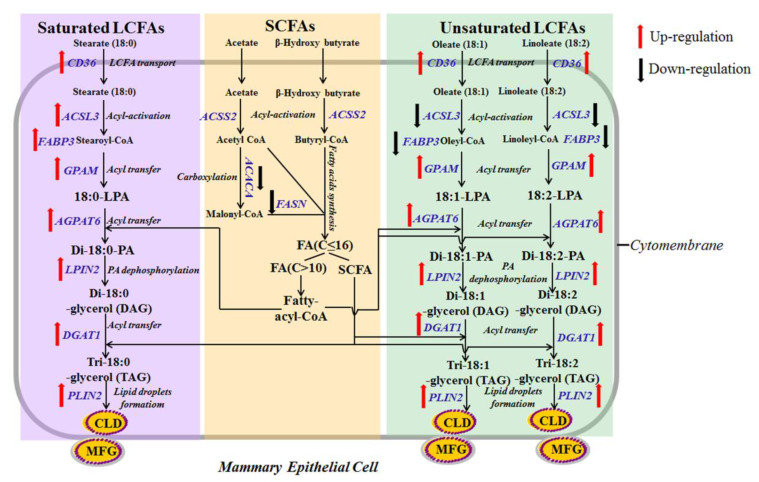
Scheme summarizing interrelationships among cellular pathways regulating lipid synthesis by stearate, oleate and linoleate in pMECs. The 18-carbon fatty acids (stearate, oleate and linoleate) (C18:X) enhanced the uptake of exogenous LCFA, TAG synthesis and lipid droplet formation. Uptake of LCFA in pMECs was enhanced by 18-carbon fatty acids (C18:X) through activating transport proteins (mainly CD36). Cytosolic C18:X is converted into its activated form (18:X-CoA) with the help of ACSL. Cytosolic 18:X-CoA is transported to the endoplasmic reticulum membrane by FABP and esterified there to glycerol-3-phosphate to produce 18:X-lysophosphatidic acid (18:X-LPA) by GPAM. Interestingly, the mRNA expressions of ACSL3 and FABP3 were different with 18-carbon fatty acids. Cellular ACSL3 and FABP3 mRNA expressions were increased by stearate but decreased by oleate or linoleate. In the endoplasmic reticulum, the addition of a second 18:X-CoA produces di-18:X-phosphatidic acid (di-18:X-PA), and di-18:X-PA can be hydrolyzed with LPIN to form a di-18:X-glycerol (DAG). The sn-3 position of DAG is then acylated to form TAG by DGAT. Newly formed TAG forms a cytoplasmic lipid droplet in the ER membrane via incorporation. Then, the cytoplasmic lipid droplet is transported to the apical membrane, and eventually released. The 18-carbon fatty acids suppressed the fatty acid de novo synthesis. In mammary cell, short- and medium-chain fatty acids (almost all C4:0–C14:0 and approximately 50% of palmitic acid) were highly dependent on the de novo synthesis. A series of cytosolic enzymes are required to facilitate this process, of which FASN and ACACA are considered the crucial enzymes of cellular fatty acid de novo synthesis in the porcine mammary gland. ACACA carboxylates acetyl-CoA to form malonyl-CoA, which is further converted by FASN to fatty acids (C ≤ 16). Then, the synthesized fatty acids participate in TAG formation in endoplasmic reticulum.

**Table 1 ijms-22-01294-t001:** Effect of stearate on mRNA expression of genes involved in lipid synthesis in pMECs ^1.^.

Gene	Stearate Concentration (μM)							SEM	*p*-Value		
	0	25	50	100	200	400	600		Stearate	Linear	Quadratic
Fatty acid uptake and import into cells											
*CD36*	1.00 ± 0.47 ^c^	2.74 ± 0.53 ^bc^	4.66 ± 3.20 ^bc^	6.84 ± 2.01 ^bc^	16.56 ± 10.00 ^ab^	25.79 ± 14.47 ^a^	16.17 ± 7.89 ^ab^	1.62	<0.01	<0.001	0.90
Fatty acid activation and intra-cellular transport											
*ACSL3*	1.00 ± 0.11 ^c^	1.14 ± 0.14 ^c^	1.46 ± 0.24 ^bc^	2.43 ± 1.14 ^ab^	2.04 ± 0.27 ^ab^	2.79 ± 1.12 ^a^	2.02 ± 0.42 ^ab^	0.14	<0.01	<0.01	0.16
*FABP3*	1.00 ± 0.24 ^e^	2.30 ± 0.28 ^d^	3.50 ± 1.35 ^cd^	5.18 ± 0.20 ^ab^	6.05 ± 0.89 ^a^	5.95 ± 0.73 ^a^	4.65 ± 0.11 ^bc^	0.15	<0.0001	<0.0001	<0.0001
Fatty acid de novo synthesis and desaturation											
*ACACA*	1.00 ± 0.03 ^b^	1.36 ± 0.37 ^a^	0.77 ± 0.15 ^c^	0.52 ± 0.04 ^de^	0.67 ± 0.01 ^cd^	0.67 ± 0.04 ^cd^	0.42 ± 0.06 ^e^	0.03	<0.0001	<0.0001	0.39
*FASN*	1.00 ± 0.31 ^a^	0.26 ± 0.17 ^cd^	0.13 ± 0.02 ^de^	0.06 ± 0.01 ^f^	0.38 ± 0.20 ^bc^	0.09 ± 0.05 ^ef^	0.52 ± 0.14 ^ab^	0.04	<0.0001	<0.01	<0.0001
*SCD*	1.00 ± 0.11 ^bc^	1.40 ± 0.32 ^b^	0.65 ± 0.16 ^d^	0.41 ± 0.01 ^e^	0.84 ± 0.10 ^cd^	3.08 ± 1.07 ^a^	2.33 ± 0.51 ^a^	0.10	<0.001	<0.001	<0.001
TAG synthesis and lipid droplet formation											
*GPAM*	1.00 ± 0.06 ^d^	1.27 ± 0.35 ^cd^	1.64 ± 0.31 ^bc^	1.57 ± 0.26 ^c^	1.60 ± 0.08 ^bc^	3.19 ± 1.18 ^a^	2.42 ± 0.17 ^ab^	0.11	<0.001	<0.001	0.69
*AGPAT1*	1.00 ± 0.24 ^a^	0.87 ± 0.19 ^ab^	0.75 ± 0.08 ^bc^	0.41 ± 0.04 ^d^	0.60 ± 0.09 ^cd^	0.96 ± 0.11 ^ab^	0.90 ± 0.11 ^ab^	0.03	<0.001	0.44	<0.0001
*AGPAT6*	1.00 ± 0.15 ^b^	1.32 ± 0.16 ^b^	1.29 ± 0.17 ^b^	1.20 ± 0.13 ^b^	1.24 ± 0.06 ^b^	1.82 ± 0.17 ^a^	2.04 ± 0.33 ^a^	0.04	<0.0001	<0.0001	0.01
*LPIN1*	1.00 ± 0.12 ^ab^	0.69 ± 0.18 ^bc^	0.47 ± 0.08 ^c^	0.33 ± 0.06 ^c^	0.94 ± 0.32 ^b^	0.97 ± 0.21 ^ab^	1.37 ± 0.38 ^a^	0.05	<0.01	<0.01	<0.001
*LPIN2*	1.00 ± 0.13 ^bc^	1.36 ± 0.09 ^b^	1.06 ± 0.05 ^bc^	0.58 ± 0.14 ^d^	0.79 ± 0.09 ^c^	2.44 ± 0.67 ^a^	2.67 ± 0.54 ^a^	0.07	<0.0001	<0.0001	<0.0001
*DGAT1*	1.00 ± 0.01 ^c^	1.70 ± 0.05 ^b^	2.39 ± 0.50 ^a^	2.63 ± 0.50 ^a^	2.79 ± 0.04 ^a^	3.03 ± 0.52 ^a^	2.63 ± 0.09 ^a^	0.07	<0.0001	<0.0001	<0.001
*PLIN2*	1.00 ± 0.20 ^e^	5.13 ± 2.33 ^d^	6.49 ± 0.34 ^cd^	13.14 ± 4.35 ^bc^	21.08 ± 8.63 ^ab^	49.75 ± 33.28 ^a^	16.74 ± 3.64 ^ab^	2.88	<0.0001	<0.01	0.52
Regulation of transcription											
*SREBP1*	1.00 ± 0.29 ^a^	0.38 ± 0.20 ^c^	0.38 ± 0.21 ^c^	0.14 ± 0.03 ^d^	0.42 ± 0.10 ^bc^	0.31 ± 0.15 ^cd^	0.71 ± 0.03 ^ab^	0.04	<0.001	0.07	<0.0001
*SCAP*	1.00 ± 0.47 ^ab^	1.34 ± 0.18 ^a^	1.06 ± 0.20 ^a^	0.68 ± 0.19 ^bc^	0.53 ± 0.12 ^c^	0.57 ± 0.09 ^c^	0.58 ± 0.09 ^c^	0.05	<0.01	<0.001	0.71
*PPARa*	1.00 ± 0.19 ^a^	0.97 ± 0.48 ^a^	0.74 ± 0.08 ^ab^	0.74 ± 0.06 ^ab^	0.55 ± 0.04 ^b^	0.82 ± 0.08 ^ab^	0.86 ± 0.12 ^ab^	0.04	0.14	0.17	0.04
*PPARγ*	1.00 ± 0.01 ^c^	1.36 ± 0.05 ^c^	2.01 ± 0.13 ^c^	3.63 ± 0.57 ^ab^	3.74 ± 0.23 ^ab^	3.94 ± 1.73 ^a^	2.27 ± 0.85 ^bc^	0.17	<0.01	<0.01	<0.01
*INSIG1*	1.00 ± 0.09 ^b^	0.99 ± 0.14 ^b^	0.55 ± 0.13 ^c^	0.41 ± 0.06 ^c^	0.87 ± 0.05 ^b^	1.51 ± 0.11 ^a^	1.50 ± 0.09 ^a^	0.02	<0.0001	<0.0001	<0.0001

^1^ The data are expressed as the mean ± SD (*n* = 3). Data were analyzed using general linear model procedure of SAS software (SAS 9.0) as a completely randomized design. Polynomial contrasts were used to evaluate linear and quadratic effects of stearate on the various response criteria. One-way ANOVA test and Student–Newman–Keuls test were used to evaluate the differences amongst groups in each experiment. ^a,b,c,d,e,f^ Different letters indicate statistical significance between different concentrations of stearate treatment groups (*p* < 0.05).

**Table 2 ijms-22-01294-t002:** Effect of oleate on mRNA expression of genes involved in lipid synthesis in pMECs ^1^.

Gene	Oleate Concentration (μM)							SEM	*p*-Value		
	0	25	50	100	200	400	600		Oleate	Linear	Quadratic
Fatty acid uptake and import into cells											
*CD36*	1.00 ± 0.01 ^e^	1.32 ± 0.11 ^d^	1.56 ± 0.24 ^c^	1.54 ± 0.04 ^c^	3.00 ± 0.29 ^b^	3.00 ± 0.25 ^b^	5.11 ± 0.15 ^a^	0.04	<0.0001	<0.0001	<0.0001
Fatty acid activation and intra-cellular transport											
*FABP3*	1.00 ± 0.09 ^ab^	0.98 ± 0.13 ^abc^	1.04 ± 0.33 ^ab^	0.64 ± 0.16 ^c^	0.76 ± 0.17 ^bc^	0.85 ± 0.16 ^abc^	1.19 ± 0.19 ^a^	0.04	0.05	0.95	<0.01
*ACSL3*	1.00 ± 0.27 ^bc^	1.29 ± 0.02 ^a^	1.25 ± 0.08 ^ab^	0.57 ± 0.14 ^d^	0.62 ± 0.13 ^d^	0.66 ± 0.08 ^d^	0.89 ± 0.06 ^c^	0.03	<0.0001	<0.0001	0.04
Fatty acid de novo synthesis											
*ACACA*	1.00 ± 0.11	1.15 ± 0.13	0.97 ± 0.34	0.56 ± 0.26	0.90 ± 0.10	0.82 ± 0.24	1.10 ± 0.20	0.05	0.06	0.49	0.03
*FASN*	1.00 ± 0.18 ^a^	0.76 ± 0.12 ^b^	0.52 ± 0.11 ^c^	0.39 ± 0.07 ^cd^	0.34 ± 0.05 ^d^	0.31 ± 0.03 ^d^	0.38 ± 0.10 ^cd^	0.02	<0.0001	<0.0001	<0.001
TAG synthesis and lipid droplet formation											
*GPAM*	1.00 ± 0.03 ^b^	1.39 ± 0.03 ^a^	1.33 ± 0.18 ^a^	1.23 ± 0.16 ^a^	1.20 ± 0.08 ^a^	1.22 ± 0.06 ^a^	1.28 ± 0.11 ^a^	0.02	0.02	0.31	0.07
*AGPAT1*	1.00 ± 0.05	1.27 ± 0.28	1.16 ± 0.21	0.91 ± 0.13	0.95 ± 0.05	1.12 ± 0.27	1.14 ± 0.19	0.04	0.26	0.86	0.45
*AGPAT6*	1.00 ± 0.06 ^bc^	1.15 ± 0.07 ^b^	1.38 ± 0.11 ^a^	0.99 ± 0.13 ^bc^	0.94 ± 0.10 ^c^	0.99 ± 0.06 ^bc^	1.00 ± 0.04 ^bc^	0.02	<0.001	0.02	<0.01
*LPIN1*	1.00 ± 0.32 ^a^	0.77 ± 0.09 ^ab^	0.69 ± 0.19 ^ab^	0.45 ± 0.19 ^b^	0.56 ± 0.11 ^b^	0.48 ± 0.17 ^b^	0.67 ± 0.15 ^ab^	0.04	0.04	<0.01	0.01
*LPIN2*	1.00 ± 0.11 ^bc^	1.56 ± 0.36 ^a^	1.25 ± 0.24 ^abc^	1.10 ± 0.11 ^bc^	0.93 ± 0.10 ^c^	1.25 ± 0.25 ^abc^	1.32 ± 0.20 ^ab^	0.05	0.04	1.00	0.56
*DGAT1*	1.00 ± 0.12 ^c^	1.23 ± 0.16 ^bc^	2.31 ± 1.14 ^a^	1.67 ± 0.07 ^ab^	1.89 ± 0.05 ^a^	1.79 ± 0.25 ^ab^	1.67 ± 0.34 ^ab^	0.10	<0.01	0.08	0.03
*PLIN2*	1.00 ± 0.21 ^d^	1.33 ± 0.41 ^d^	2.25 ± 0.34 ^c^	2.16 ± 0.52 ^c^	6.45 ± 1.96 ^b^	8.40 ± 4.99 ^b^	17.55 ± 3.64 ^a^	0.54	<0.0001	<0.0001	<0.001
Regulation of transcription											
*SREBP1*	1.00 ± 0.32 ^a^	0.90 ± 0.19 ^a^	0.57 ± 0.19 ^b^	0.44 ± 0.03 ^bc^	0.31 ± 0.04 ^c^	0.29 ± 0.03 ^c^	0.39 ± 0.13 ^bc^	0.04	<0.001	<0.0001	0.01
*SCAP*	1.00 ± 0.23 ^ab^	0.69 ± 0.12 ^c^	0.68 ± 0.24 ^c^	0.72 ± 0.09 ^bc^	0.67 ± 0.04 ^c^	0.66 ± 0.11 ^c^	1.18 ± 0.25 ^a^	0.04	0.01	0.39	<0.001
*PPARa*	1.00 ± 0.10	1.13 ± 0.22	1.01 ± 0.15	1.05 ± 0.07	0.94 ± 0.04	1.04 ± 0.20	1.05 ± 0.14	0.03	0.83	0.81	0.82
*PPARγ*	1.00 ± 0.06 ^b^	1.46 ± 0.26 ^ab^	1.48 ± 0.23 ^ab^	1.60 ± 0.35 ^ab^	1.60 ± 0.06 ^ab^	1.70 ± 0.73 ^ab^	2.04 ± 0.19 ^a^	0.07	0.04	0.03	0.31
*INSIG1*	1.00 ± 0.22 ^a^	1.07 ± 0.45 ^a^	0.70 ± 0.48 ^ab^	0.30 ± 0.09 ^bc^	0.28 ± 0.02 ^bc^	0.18 ± 0.08 ^c^	0.34 ± 0.15 ^bc^	0.06	<0.01	<0.001	0.09

^1^ The data are expressed as the mean ± SD (*n* = 3). Data were analyzed using general linear model procedure of SAS software (SAS 9.0) as a completely randomized design. Polynomial contrasts were used to evaluate linear and quadratic effects of oleate on the various response criteria. One-way ANOVA test and Student–Newman–Keuls test were used to evaluate the differences amongst them. ^a,b,c,d,e^ Different letters indicate statistical significance between different concentrations of oleate treatment groups (*p* < 0.05).

**Table 3 ijms-22-01294-t003:** Effect of linoleate on mRNA expression of genes involved in lipid synthesis in pMECs ^1^.

Gene	Linoleate Concentration (μM)							SEM	*p*-Value		
	0	25	50	100	200	400	600		Linoleate	Linear	Quadratic
Fatty acid uptake and import into cells											
*CD36*	1.00 ± 0.13 ^d^	2.11 ± 1.55 ^cd^	2.71 ± 1.00 ^cd^	3.58 ± 0.58 ^c^	6.95 ± 1.08 ^b^	7.35 ± 1.61 ^b^	10.63 ± 1.15 ^a^	0.25	<0.0001	<0.0001	0.03
Fatty acid activation and intra-cellular transport											
*ACSL3*	1.00 ± 0.03 ^a^	0.91 ± 0.11 ^a^	1.05 ± 0.24 ^a^	0.18 ± 0.07 ^b^	0.30 ± 0.06 ^b^	0.27 ± 0.05 ^b^	0.27 ± 0.06 ^b^	0.02	< 0.0001	<0.0001	0.02
*FABP3*	1.00±0.14 ^b^	1.42 ± 0.08 ^a^	1.42 ± 0.17 ^a^	0.83 ± 0.12 ^bc^	0.71 ± 0.08 ^c^	0.74 ± 0.11 ^c^	0.30 ± 0.02 ^d^	0.02	<0.0001	<0.0001	<0.001
Fatty acid de novo synthesis											
*ACACA*	1.00 ± 0.14 ^a^	0.97 ± 0.36 ^ab^	1.00 ± 0.04 ^a^	0.67 ± 0.18 ^bc^	0.72 ± 0.02 ^abc^	0.69 ± 0.06 ^abc^	0.60 ± 0.05 ^c^	0.04	0.03	<0.01	0.89
*FASN*	1.00 ± 0.03 ^a^	0.61 ± 0.13 ^bc^	0.62 ± 0.08 ^b^	0.31 ± 0.18 ^d^	0.41 ± 0.12 ^cd^	0.55 ± 0.05 ^bc^	0.56 ± 0.05 ^bc^	0.02	<0.0001	<0.001	<0.0001
TAG synthesis and lipid droplet formation											
*GPAM*	1.00 ± 0.07 ^c^	1.20 ± 0.07 ^b^	1.33 ± 0.18 ^ab^	1.49 ± 0.17 ^a^	1.54 ± 0.07 ^a^	1.44 ± 0.02 ^a^	1.49 ± 0.06 ^a^	0.02	<0.01	<0.0001	<0.01
*AGPAT1*	1.00 ± 0.16	1.21 ± 0.24	1.28 ± 0.10	1.20 ± 0.06	1.27 ± 0.01	1.31 ± 0.06	1.23 ± 0.01	0.03	0.13	0.04	0.07
*AGPAT6*	1.00 ± 0.11 ^c^	1.10 ± 0.13 ^bc^	1.23 ± 0.02 ^ab^	1.23 ± 0.06 ^ab^	1.32 ± 0.14 ^a^	1.33 ± 0.15 ^a^	1.24 ± 0.06 ^ab^	0.02	0.02	<0.01	0.03
*LPIN1*	1.00 ± 0.03 ^bc^	0.76 ± 0.24 ^cd^	0.64 ± 0.19 ^de^	0.43 ± 0.09 ^e^	1.37 ± 0.20 ^a^	1.21 ± 0.07 ^ab^	1.28 ± 0.11 ^ab^	0.03	<0.0001	<0.001	<0.001
*DGAT1*	1.00 ± 0.12 ^b^	1.26 ± 0.09 ^a^	1.26 ± 0.06 ^a^	1.25 ± 0.05 ^a^	0.96 ± 0.19 ^bc^	0.90 ± 0.18 ^bc^	0.74 ± 0.07 ^c^	0.03	<0.001	0.02	<0.001
*PLIN2*	1.00 ± 0.02 ^d^	2.36 ± 0.43 ^cd^	4.14 ± 1.40 ^cd^	5.42 ± 1.01 ^c^	8.91 ± 1.74 ^b^	15.10 ± 3.21 ^a^	12.57 ± 3.24 ^a^	0.43	<0.0001	<0.0001	0.51
Regulation of transcription											
*SREBP1*	1.00 ± 0.13 ^a^	0.52 ± 0.15 ^bc^	0.57 ± 0.08 ^b^	0.38 ± 0.10 ^c^	0.36 ± 0.03 ^c^	0.50 ± 0.06 ^bc^	0.48 ± 0.05 ^bc^	0.02	<0.001	<0.0001	<0.0001
*SCAP*	1.00 ± 0.14	1.10 ± 0.21	1.18 ± 0.09	0.93 ± 0.15	1.13 ± 0.12	1.10 ± 0.03	1.13 ± 0.07	0.03	0.29	0.43	0.94
*PPARa*	1.00 ± 0.17	1.26 ± 0.24	1.23 ± 0.12	1.17 ± 0.11	1.21 ± 0.10	1.11 ± 0.09	1.17 ± 0.04	0.03	0.39	0.71	0.16
*PPARγ*	1.00 ± 0.14 ^c^	1.66 ± 0.20 ^b^	1.70 ± 0.05 ^b^	1.84 ± 0.04 ^b^	2.41 ± 0.47 ^a^	2.50 ± 0.41 ^a^	1.92 ± 0.24 ^b^	0.06	<0.001	<0.0001	<0.01
*INSIG1*	1.00 ± 0.11 ^a^	0.76 ± 0.20 ^ab^	0.54 ± 0.03 ^b^	0.32 ± 0.09 ^c^	0.31 ± 0.02 ^c^	0.25 ± 0.06 ^c^	0.35 ± 0.08 ^c^	0.02	<0.001	<0.0001	<0.0001

^1^ The data are expressed as the mean ± SD (*n* = 3). Data were analyzed using general linear model procedure of SAS software (SAS 9.0) as a completely randomized design. Polynomial contrasts were used to evaluate linear and quadratic effects of linoleate on the various response criteria. One-way ANOVA test and Student–Newman–Keuls test were used to evaluate the differences amongst groups in each experiment. ^a,b,c,d,e^ Different letters indicate statistical significance between different concentrations of linoleate treatment groups (*p* < 0.05).

## Data Availability

The data presented in this study are available from the corresponding author upon reasonable request.

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
