# Peer review of "Metabolic Transition of Milk Triacylglycerol Synthesis in Response to Varying Levels of Three 18-Carbon Fatty Acids in Porcine Mammary Epithelial Cells"

_ijms, 2021, doi:10.3390/ijms22031294_

Round 1
Reviewer 1 Report
- Authors seem to focused on the number of carbon atoms in the FA chain, while for observed dependencies mainly saturation/unsaturation degree of FA is responsible. Thus more consideration of this aspect of obtained results should be added into discussion and conclusion.
- Why Authors choose only stearic, oleic and linoleic acids for this research? What about linolenic (C18:3) and stearidonic (C18:4) acids? Also some attention should be paid to the conjugated isomers of C18:2 (CLA) and C18:3 (CLnA) due to the significant differences in their bioactive properties and in impact on lipid metabolism in comparison to their non-conjugated counterparts. Please discuss.
- Why TAG content is presented in relation to the protein (Figure 2)? Please explain.
- The quality of Figure 3 and 4 is not sufficient. Please make them larger or more visible, especially the superscripts above the bars.
- Did Authors check how the same dose of each FA influence the measured parameters? Please provide such statistical analysis as it may confirm not only the dose dependent differences but also the differences depending on the type of used FA.
Author Response
Response to Reviewer 1 Comments
Point 1: Authors seem to focused on the number of carbon atoms in the FA chain, while for observed dependencies mainly saturation/unsaturation degree of FA is responsible. Thus more consideration of this aspect of obtained results should be added into discussion and conclusion. 

Response 1: In the revised manuscript, it was discussed with saturated 18-carbon LCFAs (stearate) or unsaturated 18-carbon LCFAs (oleate or linoleate) and the lipid synthesis pathway affected by 18-carbon LCFAs were summarized in Figure 5.
Point 2: Why Authors choose only stearic, oleic and linoleic acids for this research? What about linolenic (C18:3) and stearidonic (C18:4) acids? Also some attention should be paid to the conjugated isomers of C18:2 (CLA) and C18:3 (CLnA) due to the significant differences in their bioactive properties and in impact on lipid metabolism in comparison to their non-conjugated counterparts. Please discuss.
Response 2: In our earlier study, we found that The C16:0, C18:1, C18:2(n-6) were the highest FAs concentrations in both colostrum and milk (Lv, et al., 2015, OMICS, 19(10): 602-616). On average, stearate, oleate and linoleate contents (percent of total fatty acids, wt %) were 4.5, 32, 14, respectively, which represent the largest composition of 18-carbon fatty acids in sow milk. Therefore, sterate, oleate and linoleate were chosen as typical 18-carbon fatty acids for studying their modulatory role in regulating lipid synthesis in pMECs. It is a good ideal to test the biofunctions of polyunsaturated 18-carbon fatty acids (C18:3, C18:4) in porcine mammary gland cells. For the future study, we will consider to investigate the molecular mechanism by which conjugated isomers of C18:2 (CLA) and C18:3 (CLnA) regulate lipid synthesis in pMECs.
Point 3: Why TAG content is presented in relation to the protein (Figure 2)? Please explain.
Response 3: In this study, the content of intracellular TAG was determined by normalization to the total cellular protein content of each sample according to previous method (Sun et al., Anim Sci J. 2016, 87(2):242-249; Tang et al., J Dairy Sci. 2017, 100(5):4102-4112). The protein content was assayed as a normalization coefficient factor, aiming to minimizing the effects of cell numbers among different replicate cells.
Point 4: The quality of Figure 3 and 4 is not sufficient. Please make them larger or more visible, especially the superscripts above the bars.
Response 4: In the revised manuscript, Figure 3 and 4 were edited with higher quality in an 800 pixel.
Point 5: Did Authors check how the same dose of each FA influence the measured parameters? Please provide such statistical analysis as it may confirm not only the dose dependent differences but also the differences depending on the type of used FA.
Response 5: Though there were similar tendency of TAG content, and genes and proteins related to TAG synthesis with increasing three 18-carbon fatty acids (stearate, oleate and linoleate), the three fatty acids seem to exert different dose response based on the present tables. Considering that the cell experiment for each fatty acids were conducted independently, but without the same incubation condition, it seems uncertain what’s the exactly different effects of three types of 18-carbon fatty acids on intracellular TAG synthesis on the same concentration basis. This hypothesis will be tested in our further study with same incubation condition.
Reviewer 2 Report
This manuscript examine the effects of three 18-carbon fatty acids on mammary lipogenesis and provides new findings of function of accumulated lipid droplets in mammary epithelial cells. Methods used in this study are appropriate.
There are some questions as follows.
- In Fig 2, TAG contents in pMEC cells with high concentrations (400 to 600uM) of stearate decreased compared to that at 200uM. Why?
- The genes involved in fatty acid de novo synthesis and transcription (SREBP1) were decreased by incubation with increasing concentrations of 18-carbon fatty acid. These conditions are considered to be insulin resistance. Did you examine the gene expression and transcription in pMEC cells with insulin in the culture medium.
Author Response
Response to Reviewer 2 Comments
Point 1: In Fig 2, TAG contents in pMEC cells with high concentrations (400 to 600uM) of stearate decreased compared to that at 200uM. Why? 

Response 1: It is probably because high concentration of saturated 18-carbon LCFAs (stearate) inhibited the cell viability. As shown in Fig.1, cell viability were suppressed when incubation stearate concentration was higher than100 µM, especially at 400 or 600 µM. Compared with the treatment of 200µM, treating 400 or 600 µM stearate decreased the TAG content, probably due to the suppressed cell viability at a relative higher concentration of stearate, but the intracellular TAG content was higher in 400 or 600µM than 0~100 µM.
Point 2: The genes involved in fatty acid de novo synthesis and transcription (SREBP1) were decreased by incubation with increasing concentrations of 18-carbon fatty acid. These conditions are considered to be insulin resistance. Did you examine the gene expression and transcription in pMEC cells with insulin in the culture medium.
Response 2: In this study, SREBP1, an important transcription factor in stimulating de novo synthesis. The decreased SREBP1 in response to increasing concentration of 18-carbon fatty acids represent and negative feedback regulation of fatty acids synthesis in porcine mammary gland cells.
The pancreatic hormone insulin plays a fundamental role in regulating metabolic changes that develop during the peripartum. It is reported that lipid disorders have often been associated with impaired insulin action (Hoenig and Sellke, 2010, Atherosclerosis, 211(1):260-265). Insulin resistance can lead to mobilization of adipose triacylglycerol. In dairy cows, the transition from gestation to lactation experience insulin resistance represents as maternal adaptation to partition nutrients toward the mammary gland to support milk production (Oliveira et al. J Dairy Sci. 99(11):9174-9183; Davis A.N.2017, Graduate Theses, Dissertations, and Problem Reports. 7075).
It is regretful that the genes related to insulin resistance were not examined in this study.